# Defect Localization in Metal Plates Using Vibroacoustic Modulation

**Mohammad M. Bazrafkan** *  and **Marcus Rutner** *

Institute for Metal and Composite Structures, Hamburg University of Technology, Denickestr. 17, 21073 Hamburg, Germany

* Correspondence: mohammad.bazrafkan@tuhh.de (M.M.B.); marcus.rutner@tuhh.de (M.R.); Tel.: +49-(0)40-42878-3222 (M.M.B.); +49-(0)40-42878-3022 (M.R.)

**Abstract:** This paper reviews the state-of-the-art approaches in defect localization and specifies the remaining questions and challenges. Furthermore, this study presents a novel defect localization methodology using the nonlinear interaction of primary Lamb wave modes and vibroacoustic modulation (VAM), combined with damage imaging, to address the current shortcomings of defect localization. The study investigates this methodology experimentally with respect to defect interpretation, resolution, and applicability. Two Lamb waves with high and low frequencies, one being continuous and the other a tone burst, were excited using two different piezoelectric sensors. The amplitude of the measured signal at the first sideband frequency was evaluated with a short-time Fourier transform (STFT) and used for damage imaging via the delay and sum method. This study also includes a discussion on identifying the source of nonlinearity reflected in the first sideband. The experimental measurements prove that the localization of defect nonlinearity is possible with high accuracy, without the need for a baseline measurement, and with a minimum number of sensors. Sensitivity measurements with respect to the required length of the high-frequency tone burst and the sensor arrangement were also conducted.

**Keywords:** damage localization; Lamb waves; damage imaging; short-time Fourier transform; vibroacoustic; baseline-free; piezoelectric; tone burst

## 1. Introduction

Reliable structural defect localization, particularly the localization of defects in a large component or connection and achieved with a reduced number of sensors, still constitutes a challenge in engineering. Defect localization has been investigated by many previous research groups, and the reported results demonstrate that defect localization is possible under certain conditions, using non-destructive testing (NDT) techniques, as discussed in the following article.

One common technique for localization involves using a single Lamb wave signal. Lamb waves achieve quite long propagation distances in thin plate structures, which is the main reason why linear and nonlinear Lamb wave detection techniques have attracted research attention. The term "linear" refers to amplitude and phase change, while "nonlinear" refers to a change in second and higher harmonics.

A Lamb wave is a single tone burst at a specific frequency, applied by a piezoelectric sensor or a combination of piezoelectric sensors. Simultaneously, the signal is measured using one or more receivers, where the receiver is also a piezoelectric sensor. The first measurement provides the baseline measurement, which is used to evaluate the difference between that and later measurements. A damage imaging method is employed to set the signals of receivers in comparison, revealing the damage location. Researchers have improved this algorithm by using modified damage imaging methods or using a different combination or arrangement of receivers to enhance the measured signal, as reviewed in the following studies.

Liu et al. [1] used an array of 16 piezoelectric sensors in a 4-by-4 pattern for localization. Sixteen measurements were performed. For each measurement, one of the 16 piezoelectric sensors worked as an actuator, and the other 15 piezoelectric sensors measured the signal. A continuous wavelet transform was applied to the measured signal to reduce the noise. Two images were obtained to improve the imaging quality. The first image was obtained using the total focusing method (TFM) with amplitude information, and the second was obtained using the sign coherence factor (SCF) with phase information. Two holes of different diameters and an added mass were placed on the plate to simulate different damage scenarios. The method accurately located the damaged areas with one set of measurements. Obviously, all damages were of a linear elastic nature; hence, we do not provide a proof-of-concept for detecting the material nonlinearity generated by a crack.

Promising results concerning damage localization were achieved by Lu et al. [2] using an $S_0$-mode Lamb wave excitation process, whereby the damage was idealized with a bonded mass. First, 11 piezoelectric sensors were placed on a plate. Five piezoelectric sensors were then selected randomly for the application of the localization algorithm. Furthermore, a boundary coefficient was introduced to reduce the effect of the signal when reflected back by the boundaries. Both the ellipse and hyperbola paths were applied for damage imaging in the context of the delay and sum method, to improve the accuracy of the algorithm. The findings proved that damage localization is successful with a bonded mass; however, the question remains as to whether this approach is able to detect real cracks. Furthermore, this suggested methodology requires a reference measurement.

Qiu et al. [3] introduced a novel baseline-free algorithm for damage localization using Lamb waves. The first $S_0$-mode burst arriving at the receivers was normalized within a compensation algorithm and was then used as the baseline signal. The compensation algorithm was tested both numerically and experimentally. The defect in the numerical simulation was in the form of a crack, whereas in the experiment, it was idealized as two holes of different diameters. Via a numerical assessment, the angle of the crack between the signal direction and crack path, the crack length, and their effect on signal amplitude modulation were investigated. The signal amplitude decreased sharply when the angle of damage was in the range of 15 to 60 degrees. In addition, the amplitude of the signal decreased with the increasing crack length. In the experimental study, the compensation algorithm was used for damage localization. Twelve piezoelectric sensors were placed on the plate, where one piezoelectric sensor worked as the actuator and the remaining sensors measured the signal. A probability imaging technique (PIT) was used to localize the damage, which was idealized by two holes of 2 mm and 4 mm in diameter in the plate. This baseline-free measurement algorithm located the damage in both cases. The accuracy increased with the larger-diameter hole. Based on this finding, Lamb wave defect localization is possible using a baseline-free algorithm; however, this approach requires a significant number of piezoelectric sensors and a high number of measurements to be taken between any pair of piezoelectric sensors for successful localization.

In summary, while the Lamb wave approach can localize cracks in a numerical model, the literature reviewed by the authors does not provide any successful experimental localization of a real-world crack using the Lamb wave. Further drawbacks of the known approaches are the need for a large number of sensors and the requirement regarding multiple measurements.

Another damage detection technique is vibroacoustic modulation (VAM), which was first introduced by Donskoy and Sutin [4]. In VAM, a high-frequency signal is generated by a piezoelectric sensor, whereas a shaker or loading frame generates a low-frequency signal. When both high- and low-frequency signals are produced by piezoelectric sensors, the method is referred to as nonlinear Lamb wave mixing [5–8]. VAM is employed in the literature in two different ways: in the first, at least one of the signals is applied as a tone burst waveform [9], and in the second, both high- and low-frequency signals are continuous waveforms, generating a steady-state condition targeting damage detection [10–12]. Recent studies show that the sensitivity of nonlinear Lamb wave mixing/VAM to minor damage

such as fatigue-induced macrocracks is higher than that of linear Lamb waves [5,6,13]. Furthermore, VAM also shows potential in terms of crack localization, which was first mentioned theoretically by Donskoy [4] but was not further explored experimentally. The following paragraphs review the published approaches regarding Lamb wave mixing/VAM.

Li et al. [9] investigated the localization of a 15-mm crack with a pattern of six sensors and six receivers, all acting as piezoelectric sensors, attached along both sides of a rectangular plate. A low-frequency continuous signal and a high-frequency tone burst signal of 600 μs in length were excited by one sender. The six receivers on the opposite side measured the signal simultaneously. A short-time Fourier transform with a window length of 600 μs was used to evaluate the measured signal. The signal amplitude at the sideband frequency was higher when the defect was positioned along the path between the sender and receiver. A probability damage-imaging method was used for localization in this test, in which defect localization was possible with an acceptable level of accuracy. However, successful measurement requires an array of sensors and, further, the defect being positioned within the array of piezoelectric sensors.

Li et al. [6] investigated the mixing conditions necessary for Lamb wave mixing. High and low frequencies not only need to satisfy the mixing condition (Equation (1)) but also should be low enough to excite only the $S_0$- and $A_0$-modes.

$$\frac{\omega_{S_0}}{\omega_{A_0}} = \frac{2\kappa}{\kappa + 1} \, , \; \kappa = \frac{C_{S_0}}{C_{A_0}} \tag{1}$$

In Equation (1), $\omega_{S_0}$ and $\omega_{A_0}$ refer to the frequencies of the $S_0$ and $A_0$ signal, respectively. $C_{S_0}$ and $C_{A_0}$ are the respective phase velocities of a wave. An $A_0$-mode wave at a frequency equal to the first sideband frequency, $\omega - \Omega$, is generated by a pair of $S_0$- and $A_0$-mode waves only when the mixing condition is satisfied, and the high- and low-frequency signals meet each other at the same time at the defect location. Here, $\omega$ and $\Omega$ are the high and low frequencies, respectively. This new $A_0$-mode signal propagates in the opposite direction to the primary transverse wave and thereby reveals the damage zone. A numerical simulation was used to test this condition, and the results showed that the waveform of the mixing wave changes significantly from a diamond shape to tone burst trains with an increase in frequency deviation.

Pieczonka et al. [14] used harmonic and sideband image mapping by applying the vibroacoustic modulation (VAM) technique. Low- and high-frequency signals were excited by two different piezoelectric sensors and a laser vibrometer measured the signal. Three methods, i.e., vibrothermography, higher harmonics imaging, and the sideband imaging method, were used for defect localization within a comparative study. In the vibrothermography method, a tone burst signal was excited by a piezoelectric sensor, while a thermographic camera measured the surface temperature of the plate. The surface temperature distribution indicated those areas where energy was converted to heat. The area adjacent to the crack revealed a higher temperature than the more remote locations on the plate. In the higher harmonics imaging method, a single frequency signal was excited by a piezoelectric sensor, and the amplitude of the measured signal at the first harmonic was used for localization. In the sideband imaging method, low- and high-frequency signals were excited by piezoelectric sensors. The sideband density, which is defined as the mean value of the first sideband amplitude, was used for damage imaging. Pieczonka et al. concluded that the sideband imaging method produced more accurate results than the higher harmonics imaging method.

In another study, the Lamb wave mixing method was applied for localizing the defect by evaluating the higher harmonics generated by a defect in a thin plate, both numerically and experimentally [7]. Two signals at different frequencies were excited; a time delay ensured that the two signals arrived at the defect simultaneously, and the Lamb wave mixing took place at the defect zone. The results showed that the higher harmonics and sideband frequency amplitudes were more pronounced in the defect zone. However, in

order to find the location of the defect in a plate-like component, this approach needs to be repeated for any potential defect zone in a checkerboard pattern.

Karve and Mahadevan [10] used VAM and a binary damage index for damage localization in a numerical concrete model. Low- and high-frequency signals were excited on the top side of the model, and the signals were measured using sensors that were placed on the bottom side of the model. The sum of the first sideband amplitude was used as a damage index for localization. The simulation results showed that the sensor's proximity to the damage affects the defect localization process. The sensor closer to the damage indicated a higher damage index value than the sensor positioned further away from the damage location. The damage index was evaluated for all the sensors in the damage localization process.

The interaction of an ultrasound signal and two types of cracks, an inner defect and a surface defect, with different lengths, widths, angles, and numbers, were investigated numerically by Zhan et al. [8] in different scenarios. The simulation results show that the nonlinear phenomenon strongly depends on the length and width of the cracks. The nonlinearity increased exponentially by increasing the length of both cracks; however, it decreased with increasing crack width. With respect to the angle between the crack direction and the propagation path, the study indicates that the nonlinearity decreases with an increasing angle.

Aslam et al. [15] numerically investigated a new $A_0/S_0$ mode signal generated by a defect. Two signals at different frequencies were excited to study the Lamb wave mixing process. The Lamb mixing zone was controlled by a time delay between the signals. A damage index (DI) was used to compare the results. The results showed that the DI increases and reduces with increased crack length and width, respectively.

The Lamb wave frequency-mixing method for localizing a crack in a thin plate, using a 2D finite element model (2000 mm $\times$ 2 mm), was introduced by Wang et al. [16]. An $A_0$-mode low-frequency tone burst signal and an $S_0$-mode high-frequency tone burst signal were excited by two piezoelectric sensors on the same side of the sample. A new $A_0$-mode signal was generated by the defect when both tone bursts arrived at the same time at the location of the defect. This simultaneous arrival at the defect location was controlled by a time delay in one of the signals. The newly generated $A_0$-mode signal at the frequency $\omega - \Omega$ propagated in the opposite direction and was measured by the receivers. The generated $A_0$-mode signal at $\omega - \Omega$ was extracted from the measured signal using the pulse inversion technique and a bandpass filter. The time of flight (TOF) of this extracted signal revealed the defect location within a numerical assessment range.

In summary, from the literature reviewed, the following challenges or questions regarding defect localization still exist:

1. The literature provides evidence from numerical studies on defect localization; however, there is a lack of algorithms with an experimental proof-of-concept that demonstrate the capability of localizing a real crack in a metal plate structure.
2. Which parameter optimizes the outcome of defect localization measurement? Since the study introduced herein focuses on metal structures, the defect chosen is a single crack. Various parameters are introduced in the literature for use as governing parameters to indicate the existence of defects; among them are the modulation index or damage index [4,10,12,14,15], the harmonics [7–14], and the first sideband [6,7,9,16]. The first sideband seems of particular interest since it is caused by the nonlinear behavior of the crack, and its amplitude is greater than that of the higher-order sidebands.
3. The signal measurement is affected by the proximity of adjacent piezoelectric sensors. It is important to either quantify this effect and account for it or take alternate measures, in order to avoid taking misleading measurements. For the study presented herein, the application of a narrow grid of sensors was avoided and the number of piezoelectric sensors was reduced to a minimum [17].
4. Signal nonlinearity can have various sources, such as structure-related initial, boundary-related, geometry-related, measuring instrument-related, and defect-related nonlin-

earity sources, to name a few. How do we make sure that the localization algorithm leads to the location of the crack when working in the presence of other sources of nonlinearity?

5. The applied signal duration becomes an important parameter to avoid signal overlapping. How short a signal duration can we select that will still supply information as to the defect location?

6. Can defect localization using VAM/ Lamb wave frequency-mixing be achieved when the defect is outside the arrangement of piezoelectric sensors?

As mentioned in the literature [6–8,15], the nonlinear Lamb wave mixing/VAM technique has a higher potential to reveal particularly small defects compared to a single Lamb wave approach, which can be explained by the ability to sense nonlinear signals. The single Lamb wave approach uses a single signal of one selected frequency. Then, an evolving defect modulates this single signal, causing a linear modulation. Conversely, the nonlinear Lamb wave mixing/VAM technique uses two signals of two different frequencies. An evolving defect, such as a crack, causes a modulation by using these two signals, making it a nonlinear modulation [4,6–10,12,14–16]. In the approach introduced herein, the VAM technique is combined with the delay and sum (DAS) damage imaging method for localizing a defect. This is achieved by tracking the source of the first sideband amplitude in a plate-like structure for the first time and then addressing the challenges listed above in points 1–6.

## 2. The Defect Localization Methodology

Regarding VAM theory, when two signals at low ($\Omega$) and high ($\omega$) frequencies are excited simultaneously, signals at $\omega \pm n \cdot \Omega$ are generated, which are known as sidebands. A source of nonlinearity, such as a defect evolution, generates these sidebands, as shown in Figure 1a. The block diagram of the VAM theory is illustrated in Figure 1b. $X_\omega(A_\omega, \omega)$ and $X_\Omega(A_\Omega, \Omega)$ represent the respective signals with amplitude and frequency information.

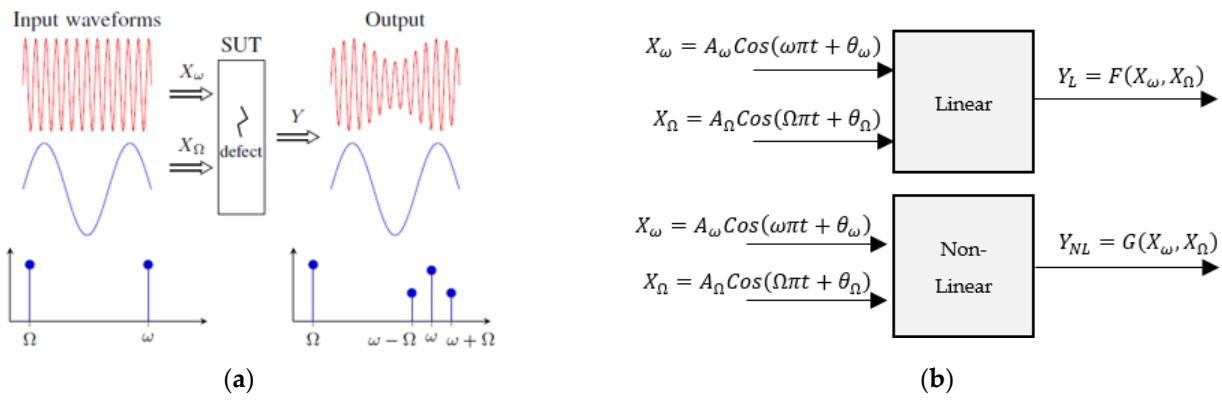

**(a)**             **(b)**

**Figure 1.** (**a**) The VAM theory—the generation of the signal at the sideband frequencies $\omega \pm n\,\Omega$, in the presence of the defect; (**b**) the block diagram of the VAM theory.

The outputs of the linear and non-linear samples are expressed by Equations (2) and (3) [4,9,12,18]:

$$Y_L = F(X_\omega, X_\Omega) = \alpha_{1L} A_\omega Cos(\omega \pi t + \theta_1) + \beta_{1L} A_\Omega Cos(\Omega \pi t + \theta_2) \tag{2}$$

$$\begin{aligned} Y_{NL} = G(X_\omega, X_\Omega) = \; & \alpha_{1NL} A_\omega Cos(\omega \pi t + \theta_3) + \beta_{1NL} A_\Omega Cos(\Omega \pi t + \theta_4) + \gamma_{1NL} A_\omega A_\Omega Cos((\omega + \Omega)\pi t + \theta_5) + \eta_{1NL} A_\omega A_\Omega Cos((\omega - \Omega)\pi t + \theta_6) \\ & + other\; harmonics(\omega, \Omega, \omega \pm \Omega), \end{aligned} \tag{3}$$

where $\alpha_{1L}$ and $\beta_{1L}$ are unknown parameters that depend on linear effects, e.g., material linearity, boundary effects, and geometry, among others. Furthermore, $\alpha_{1LNL}$, $\beta_{1LNL}$, $\gamma_{1LNL}$, and $\eta_{1LNL}$ are unknown parameters that depend on nonlinear effects, e.g., crack nonlinearity. The variable $\theta_i$ in Equations (2) and (3) represents a phase shift at different

frequencies. The suggested localization methodology tracks the signal at the first sideband frequency, $\omega + \Omega$, in order to find the source of the nonlinearity.

As mentioned before, establishing the steady-state condition is not suitable for defect localization, which requires at least one of the high- or low-frequency signals to be excited as a tone burst signal. This article selects a low-frequency (LF) signal as an $S_0$-mode continuous signal, which is excited by two piezoelectric sensors placed on both sides of the plate, as shown in Figure 2a. As soon as the LF signal reaches a steady-state condition, an $A_0$-mode high frequency (HF) tone burst signal is excited by another pair of piezoelectric sensors attached at both sides of the plate [16,18] (Figure 2a). A double-sided tape of about 0.23 mm in thickness is used for the application of the piezoelectric sensor (thickness of about 2.2 mm) on a 1.2-mm-thick aluminum plate.

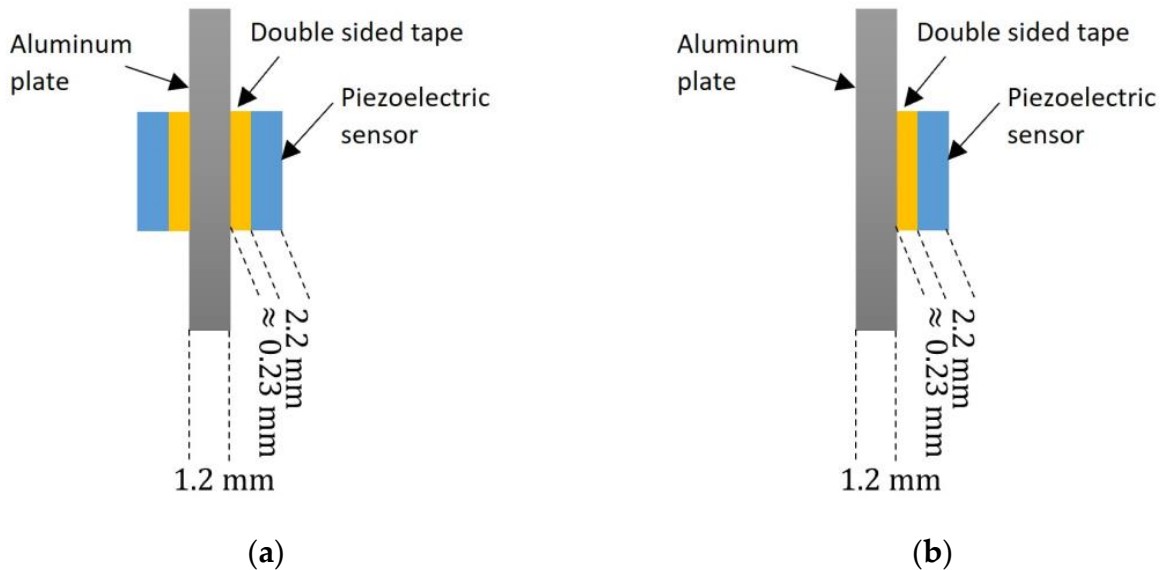

**Figure 2.** (**a**) Schematic sketch of exciting piezoelectric sensors that are attached on both sides of a plate; (**b**) the receiver attached on one side of the plate.

Four receivers in different positions are used to measure the signal. Each receiver is attached to one side of the plate, as schematically shown in Figure 2b. The time domain information of the sideband at the frequency $\omega + \Omega$, referred to as sideband amplitude (SA) in this article, is evaluated by STFT and is used for localization. In order to reduce the external nonlinearities due to measuring instrumentation [15], the SA that is evaluated from a measurement of the pure high-frequency signal ($\omega$) is subtracted from the SA evaluated from the VAM. Finally, the delay and sum (DAS) damage imaging method [19,20] was applied for localization. Figure 3 shows the suggested defect localization algorithm as a flowchart. The flowchart is split into two sub-algorithms, with the VAM measurement procedure on the left and the HF measurement procedure on the right.

### 2.1. Frequency Selection

An $S_0$-mode sinusoidal LF signal is excited by two piezoelectric sensors on both sides of the sample. The frequency of the LF signal is selected according to the eigenfrequency of the piezoelectric sensor, to achieve a high vibration amplitude. Furthermore, the necessary conditions mentioned by the authors of [18] must be fulfilled. The peak amplitude of the LF signal becomes 75 V after amplification and a duration of 400 ms to ensure that it reaches a steady-state condition. Furthermore, an $A_0$-mode tone burst of 175 kHz central frequency is chosen as the HF signal, selected according to the eigenfrequency ranges of the HF exciting piezoelectric sensors. The HF signal is excited as a Hanning windowed tone burst, with a 75 V peak amplitude and a duration ranging from 50 µs to 150 µs.

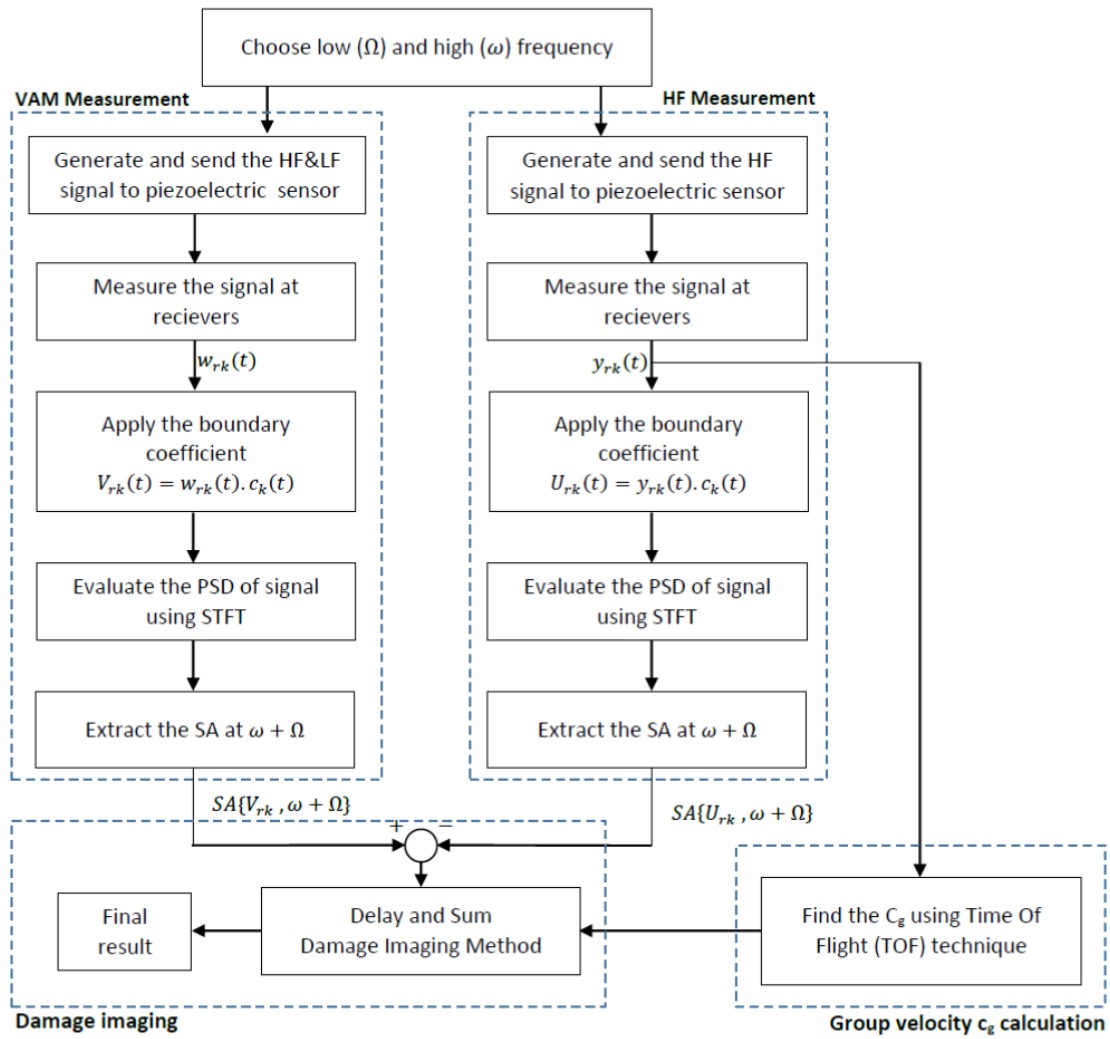

**Figure 3.** Flowchart of the defect localization algorithm.

*2.2. The VAM Measurement Procedure*

The LF continuous signal of a selected frequency is excited at time $t$ = 0 s. After a certain time delay, $t_d$, a steady-state condition is reached, and the HF tone burst signal is excited using another pair of piezoelectric sensors. Four piezoelectric sensors in different positions measure the signal $w_{rk}(t)$ simultaneously; here, $rk$ refers to receiver number $k$. The entire VAM signal-measuring process is repeated 150 times, in order to attenuate the noise level, and the averaged signal is saved.

*2.3. Boundary Coefficient*

When a tone burst signal is excited through a piezoelectric sensor on the sample, the elastic wave propagates radially; eventually, it is reflected back by the boundaries. Therefore, the receivers sense not only the first signal that arrives but also several reflected signals caused by the boundaries. In this study, the reflected signals do not overlap with the first signal that arrived. Typically, the first signal that arrives is used for localization. Two methods can be used to avoid the unwanted reflected part of the signal. One way is to cut the signal at a certain time and then use it for the localization algorithm. The other

method involves multiplying the measured signal by a boundary coefficient [2] to reduce the effect of unwanted reflected parts, as described in Equation (4).

$$z_k(t) = \begin{cases} 1 & t \le t_{km} \\ e^{-\frac{\alpha t}{t_{kj}-t_{km}}+\frac{\alpha t_{km}}{t_0-t_{km}}} & t_{km} < t < t_0 \end{cases}, \quad t_0 = \frac{a+b}{c_g} \tag{4}$$

where $c_g$ is the group velocity, $\alpha$ = 1~4 is the wave reflection strength coefficient, $a$ and $b$ refer to the side lengths, and time $t_{km} = min\{t_{kj}\}$, where $t_{kj}$ is the time taken for the wave to propagate from the actuator, be reflected at the j$^{th}$ edge, and arrive at sensor S, as shown in Figure 4. Essentially, the output signal is the product of the measured averaged signal and the boundary coefficient, $V_{rk}(t) = w_{rk}(t) \cdot z_k(t)$.

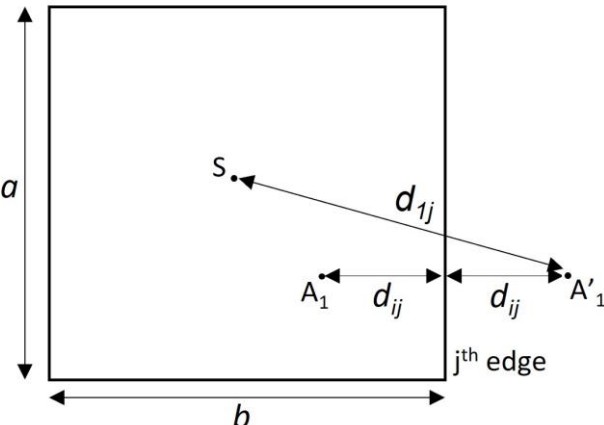

**Figure 4.** Boundary reflection wave propagating path. $A'_1$ is the mirror source at the j$^{th}$ edge.

*2.4. Short-Time Fourier Transform (STFT)*

STFT is a time-frequency transform that is used when a time history of a specific frequency is needed. The output of the STFT is the amplitude of the signal at different frequencies, which are defined by the sampling frequency and the number of samples in FFT. The parameters of the STFT in this study are as follows:

- The window function parameter is set by the Hanning window.
- The window length for each segment is selected to be equal to the duration of the HF tone burst signal, which, in our study, is within the range of 50–150 μs.
- The overlapping number is selected to increase the time accuracy of the STFT output, which, in our study, is 480 ns.
- The number of samples in FFT determines the frequency resolution, which, in our study, is 500 Hz.

The sideband amplitude (SA) at frequency $\omega + \Omega$, evaluated by STFT for the signal $u(t)$, is described by Equation (5).

$$SA\{u(t), \omega + \Omega\} = STFT\{u(t)\} \ at \ \omega + \Omega \tag{5}$$

*2.5. The HF Measurement Procedure*

The HF tone burst is excited by a piezoelectric sensor, then all four receivers measure the signals simultaneously, resulting in the measured signal $y_{rk}(t)$, as shown in Figure 3. The process is repeated 150 times and the averaged signal is saved. The measured signal, multiplied by the coefficient factor, is referred to as $U_{rk}(t) = y_{rk}(t) \cdot z_k(t)$.

*2.6. Damage Imaging Method*

Finally, the delay and sum (DAS) damage imaging method uses the results of the subtraction of $SA\{U_{rk}, \ \omega + \Omega\}$ from $SA\{V_{rk}, \ \omega + \Omega\}$ to localize a defect. The delay and

sum damage imaging method [18–22] has been modified by Michaels [19,21] and is used for defect localization. In the presence of a continuous LF signal, the HF tone burst, which is excited by piezoelectric sensors at the coordinates $(x_s, y_s)$, propagates radially within the 2D-plate sample with the group velocity $\vec{C}_{HF}$, as illustrated in Figure 5a. The HF signal reaches the damage location at the coordinates $(x_D, y_D)$ after time $t_1$, as assessed using Equation (6). At the defect location, the sideband at the frequency $\omega + \Omega$ evolves due to the local defect. This sideband signal propagates radially with the group velocity $\vec{C}_{SB}$, as shown in Figure 5a. This signal reaches the receiver located at coordinates $(x_{r1}, y_{r1})$ after time interval $t_2$, assessed in Equation (7). The total time interval $T_{SDR1}$, during which the signal propagates along the path from the actuator to the defect at coordinates $(x_D, y_D)$ and to the receiver, is provided by Equation (8). However, knowing the total time, $T_{SDR1}$, and not knowing the defect location limits the potential defect location, positioned on an ellipse. Essentially, the total time interval, $T_{SXR1}$, during which the signal propagates along the path from the actuator to a defect position at the coordinates $(x, y)$ on the ellipse and to the receiver, is provided by Equation (9).

$$t_1 = \frac{\sqrt{(x_S - x_D)^2 + (y_S - y_D)^2}}{C_{HF}}, \tag{6}$$

$$t_2 = \frac{\sqrt{(x_D - x_{r1})^2 + (y_D - y_{r1})^2}}{C_{SB}}, \tag{7}$$

$$T_{SDR1} = \frac{\sqrt{(x_S - x_D)^2 + (y_S - y_D)^2}}{C_{HF}} + \frac{\sqrt{(x_D - x_{r1})^2 + (y_D - y_{r1})^2}}{C_{SB}}, \tag{8}$$

$$T_{SXR1}(x, y) = \frac{\sqrt{(x_S - x)^2 + (y_S - y)^2}}{C_{HF}} + \frac{\sqrt{(x - x_{r1})^2 + (x - y_{r1})^2}}{C_{SB}} \tag{9}$$

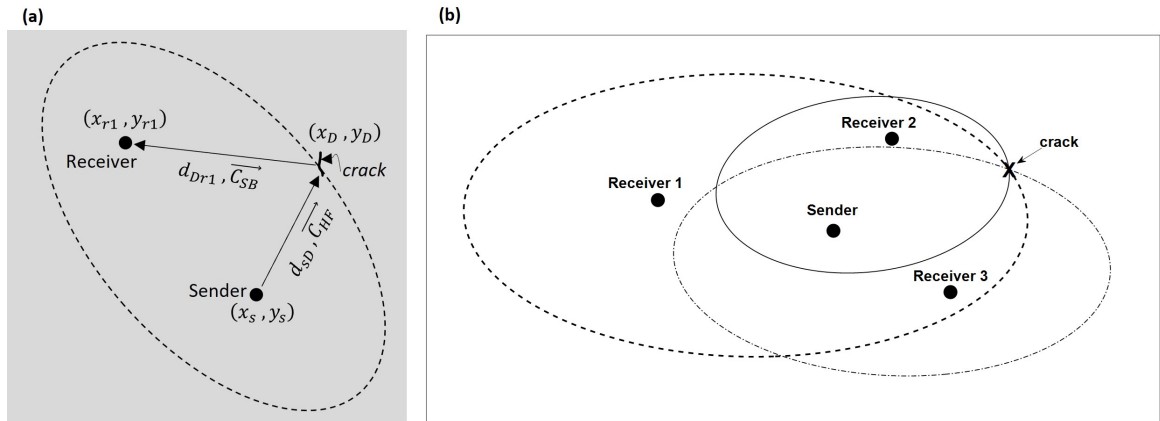

**Figure 5.** (**a**) Sensor, defect, and receiver location, and the overlaid ellipse that marks possible locations of the defect, retracted using the time interval, $T_{SDR1}$ (Equation (9)); (**b**) defect localization using three sender-receiver pairs.

### 2.7. The Group Velocity Calculation

In order to measure the group velocity at high and sideband frequencies, a tone burst signal of 50 μs in length is excited and the signal is measured by the receivers simultaneously. The group velocity of the signal is calculated using the time of flight (TOF) technique [2,18–21]. In order to evaluate the group velocity experimentally, a tone burst HF signal with a duration of 50 μs at different frequencies is excited by a piezoelectric sensor.

The signals are measured using two receivers, which are positioned at distances of 200 mm and 400 mm, respectively, as shown in Figure 6a. Figure 6b shows the experimentally measured frequency-dependent group velocity, as well as the expected theoretical group velocity.

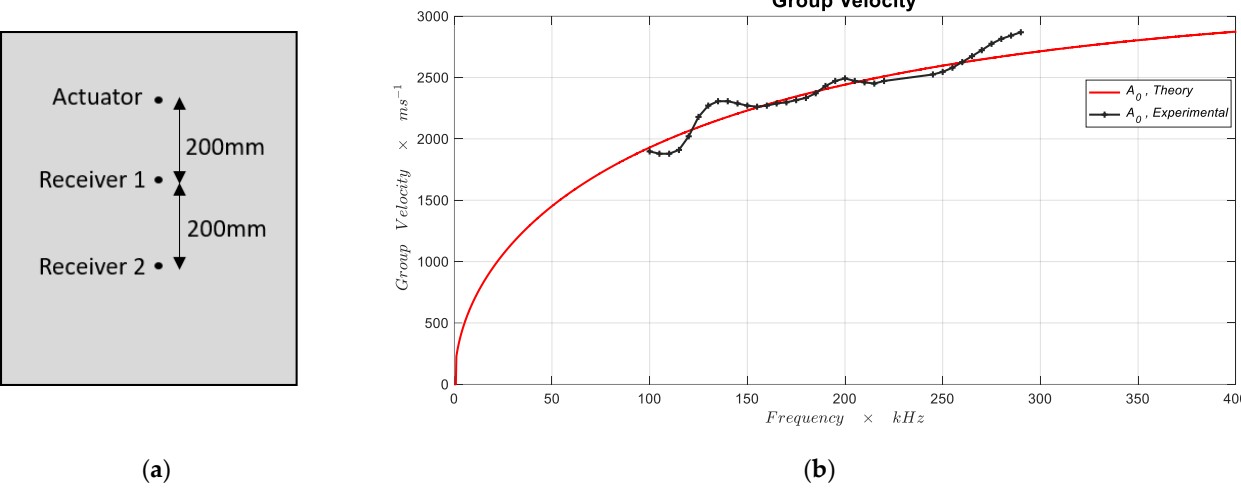

(**a**)        (**b**)

**Figure 6.** (**a**) Test setup for evaluating the group velocity; (**b**) group velocity for the different frequencies, showing the theoretical (continuous line) and experimental tests with a laser vibrometer (cross symbol).

## 3. The Specimen

The defect localization algorithm, as presented in Figure 3, was tested experimentally. The experiment used a 700 mm × 1000 mm × 1.2 mm aluminum plate specimen (AlMg3). A hole of 4.5 mm in diameter was drilled through the plate in an arbitrary position. The hole had sharp notches on two opposite sides along the *y*-axis. Their notch angle was 55°, as shown in Figure 7. A crack with a length of a = 30 mm was produced that grew out of the notch by applying cyclic loading along the short length of the metal place. The aluminum plate was installed in a test rig that held the plate specimen vertically in place using four spring connectors at the top and bottom sides, as shown in Figure 7a. Figure 7b shows close-up views of the notch, crack length, crack tip, and width.

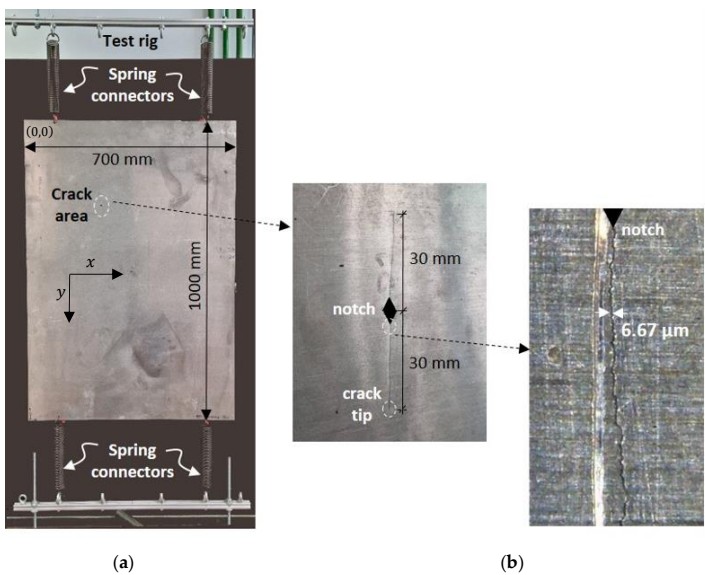

(**a**)        (**b**)

**Figure 7.** (**a**) Plate specimen installed in the test rig; (**b**) close-up view of the notch and crack (crack length a = 30 mm).

A signal generator (Agilent 33220A) generated an LF signal with a length of 400 ms at the selected frequency. The HF tone burst signal at 175 kHz, with a duration of between 50 and 150 μs, was generated by another signal generator (Keysight 33600A). The HF tone burst signal was excited after a time delay of 350 ms, after exciting the LF signal, to ensure that the LF signal had reached a steady-state condition. The LF signal, as well as the HF signal, were excited by two piezoelectric sensors (PI255 piezoceramics, PI Ceramic GmbH), which were attached on both sides of the plate (Figure 2a) to generate an $S_0$-mode LF signal and an $A_0$-mode HF signal, respectively. The signals were amplified by two WMA-320 high-voltage amplifiers, up to 75 V (peak). The piezoelectric sensors (PI255) for exciting the LF and HF signal had diameters of 26 mm and 10 mm and eigenfrequencies in the ranges of 70–80 kHz and 170–220 kHz, respectively.

The receivers, four piezoelectric sensors (PI255) with a diameter of 8 mm and an eigenfrequency range of 230–270 kHz, were attached at different positions and then used as receivers. A 4-channel RBT2004 digital oscilloscope was used for 150-times averaging and for saving the signals from the receivers. A National Instruments (NI) device (USB-6366) synchronized all equipment components and a MATLAB script automated the process.

According to the eigenfrequency range of the applied piezoelectric sensors, the LF was selected in the range of 70–95 kHz. The first sideband frequency ranges were $\omega + \Omega = 245 - 270$ kHz and $\omega - \Omega = 75 - 100$ kHz. Since the eigenfrequency range of the receivers was around 250 kHz, the first sideband frequency at $\omega + \Omega$ was selected for the localization algorithm.

Several sensor arrangements, as well as the tone burst signal durations, were used in order to investigate the sensitivity in terms of:

(a)    defect detection and
(b)    defect localization,

as is discussed in the following section.

## 4. Measurements, Findings, and Discussion

### 4.1. Defect Detection

Three receivers were positioned at a selected distance of 190 mm around an HF actuator on the plate specimen, as shown in Figure 8. The direct path from the HF actuator to Receiver 1 crossed the crack at a 90° angle, while the direct path between the HF actuator to Receivers 2 and 3, respectively, did not cross the crack. The position of the LF actuator is also shown in Figure 8. In this test, the LF actuator excited a 400 ms signal at 90 kHz, while the HF actuator excited an HF tone burst at 175 kHz with a duration of 100 μs. Theoretically, the crack was expected to affect the sideband amplitude (SA) received by Receiver 1 most prominently, compared to the signals received by Receivers 2 and 3.

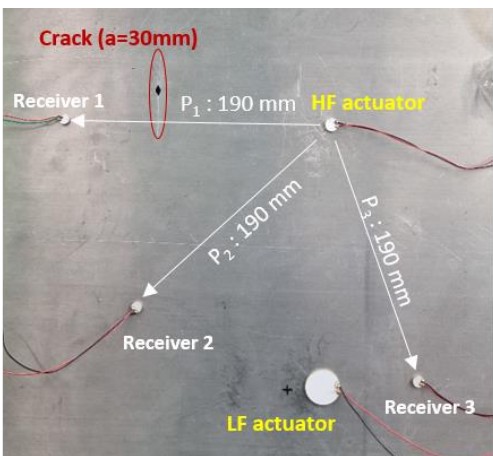

**Figure 8.** Actuator and receiver arrangement adjacent to the crack: signal sensitivity of the path crossing the defect vs. the path not crossing the defect; crack length a = 30 mm.

The measured VAM signal, $V_{rk}(t)$ (red signal), and the measured HF signal, $U_{rk}(t)$ (blue signal), recorded by Receivers 1, 2, and 3 are cross-plotted in the time domain in Figure 9. The tone burst signal caused an immediate change to the measured VAM signal, which proves that $V_{rk}(t)$ and $U_{rk}(t)$ arrived at the receiver at the same time. Furthermore, it is emphasized that the methodology proposed herein only works if the first HF tone burst that arrives is not compromised by any signal reflections from the boundaries, as shown in Figure 9a–c.

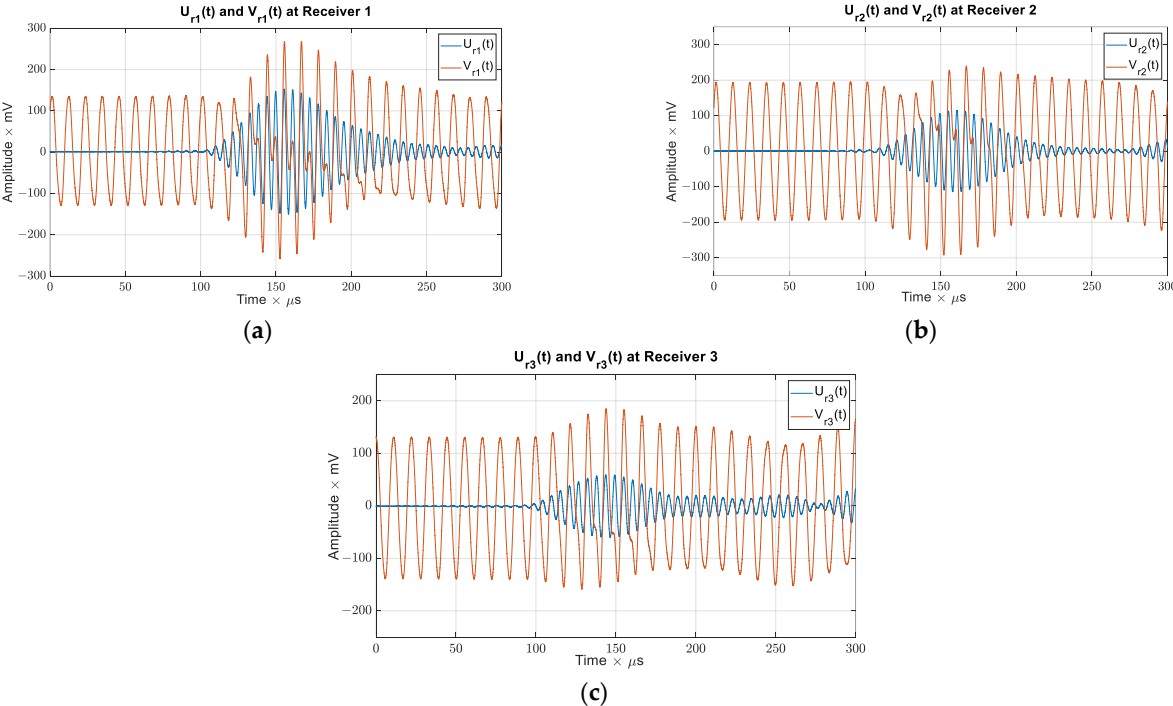

**Figure 9.** The HF signal $U_{rk}(t)$ (blue) and the VAM signal $V_{rk}(t)$ (red) for an HF tone burst with a duration of 100 μs, as sensed by Receivers 1–3, shown in (**a**–**c**), respectively.

Figure 10a represents the HF signal measured by Receiver 1, $U_{r1}(t)$, and Figure 10b shows the STFT result for the same signal. The white lines represent the high frequency at $\omega = 175$ kHz, the low frequency at $\Omega = 90$ kHz, and the first sideband frequency at $\omega + \Omega = 265$ kHz. As expected, the STFT plot reveals the highest level of energy at 175 kHz. The energy of the signal at 90 kHz and 275 kHz is almost zero. Furthermore, an increment of 300 μs is shown in Figures 10–13, in order to highlight only the first signal that arrived at the receivers.

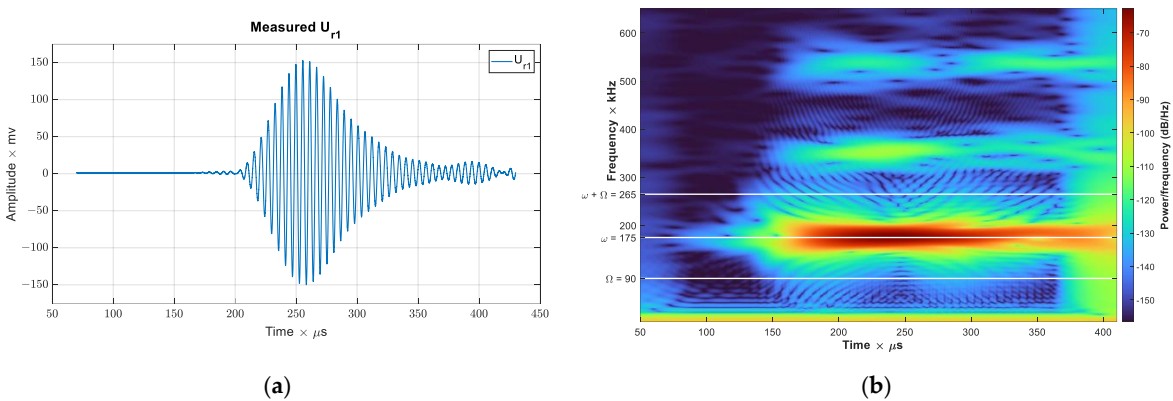

**Figure 10.** (**a**) $U_{r1}(t)$ and (**b**) STFT of $U_{r1}(t)$. White lines represent $\omega = 175$ kHz, $\Omega = 90$ kHz, and $\omega + \Omega = 265$ kHz.

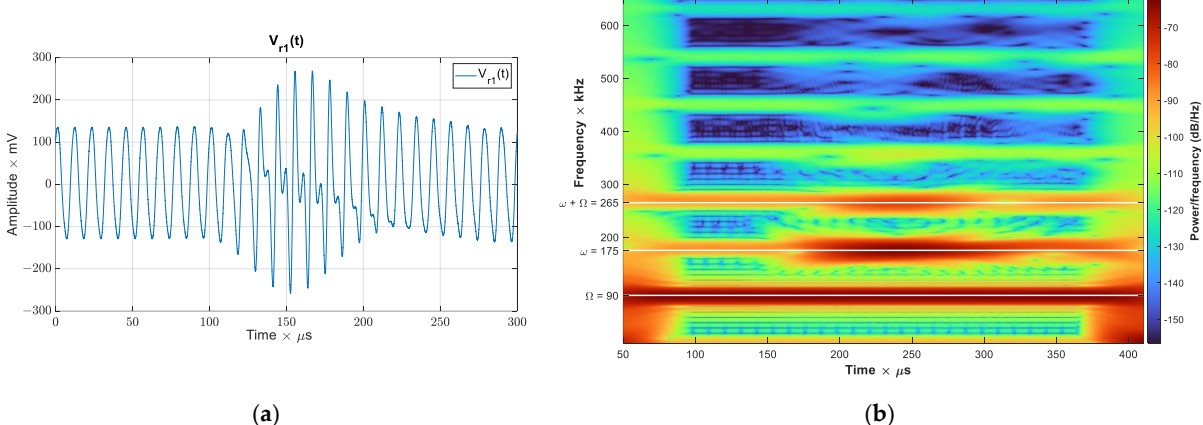

(**a**)                                                                          (**b**)

**Figure 11.** (**a**) $V_{r1}(t)$; (**b**) STFT of $V_{r1}(t)$. White lines represent $\omega = 175$ kHz, $\Omega = 90$ kHz, and $\omega + \Omega = 265$ kHz.

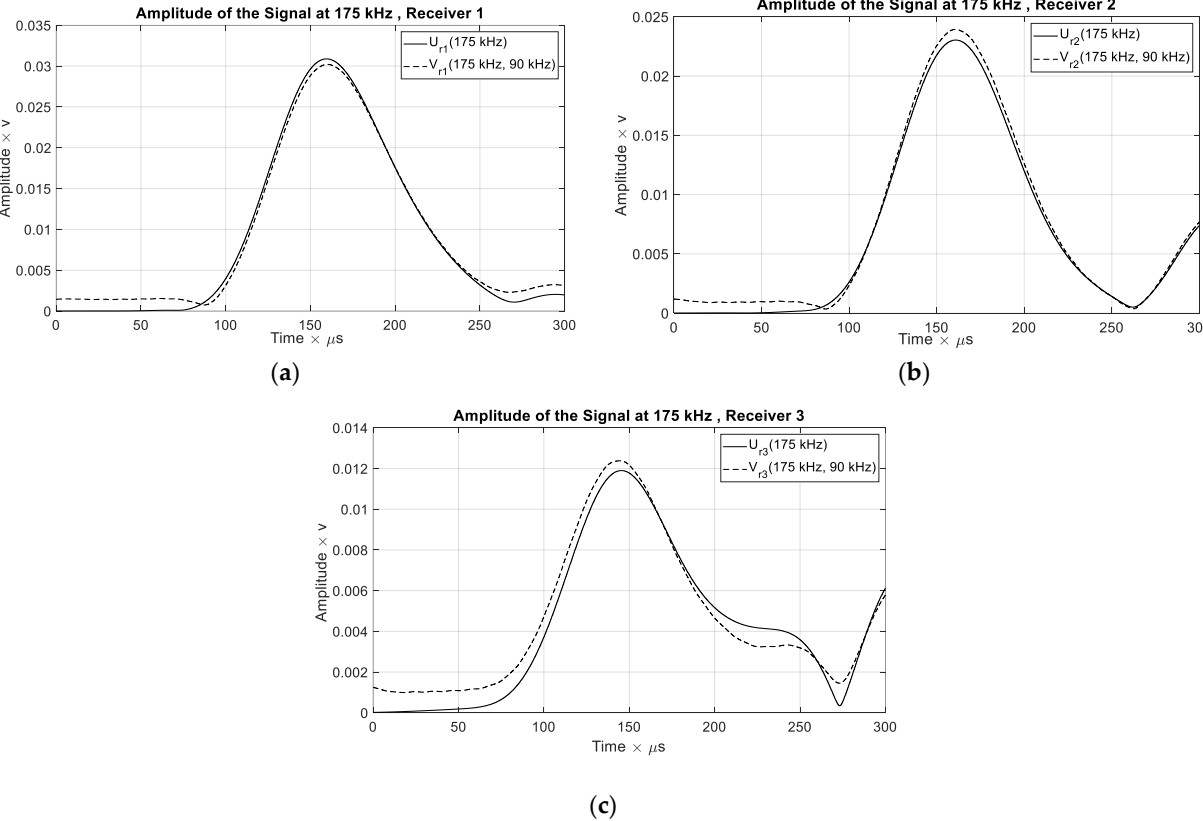

**Figure 12.** The amplitudes of $V_{rk}(t)$ and $U_{rk}(t)$ at $\omega = 175$ kHz, as sensed by Receivers 1 (**a**), 2 (**b**), and 3 (**c**).

Figure 11a shows the VAM signal received by Receiver 1, $V_{r1}(t)$. Figure 11b also provides the STFT plot of $V_{r1}(t)$. As expected, the STFT plot reveals that high energy also appears at the sideband frequency, as well as at frequencies $\omega$ and $\Omega$.

It is widely recognized that the VAM signal $V_{rk}(t)$ at frequency $\omega$ and the HF signal $U_{rk}(t)$-signal for a specific receiver show almost the same amplitude. However, the peak amplitudes differ from receiver to receiver, as shown in the time histories in Figure 12. The peak amplitudes will be used for normalizing the sideband amplitude for comparison, as explained in the next section.

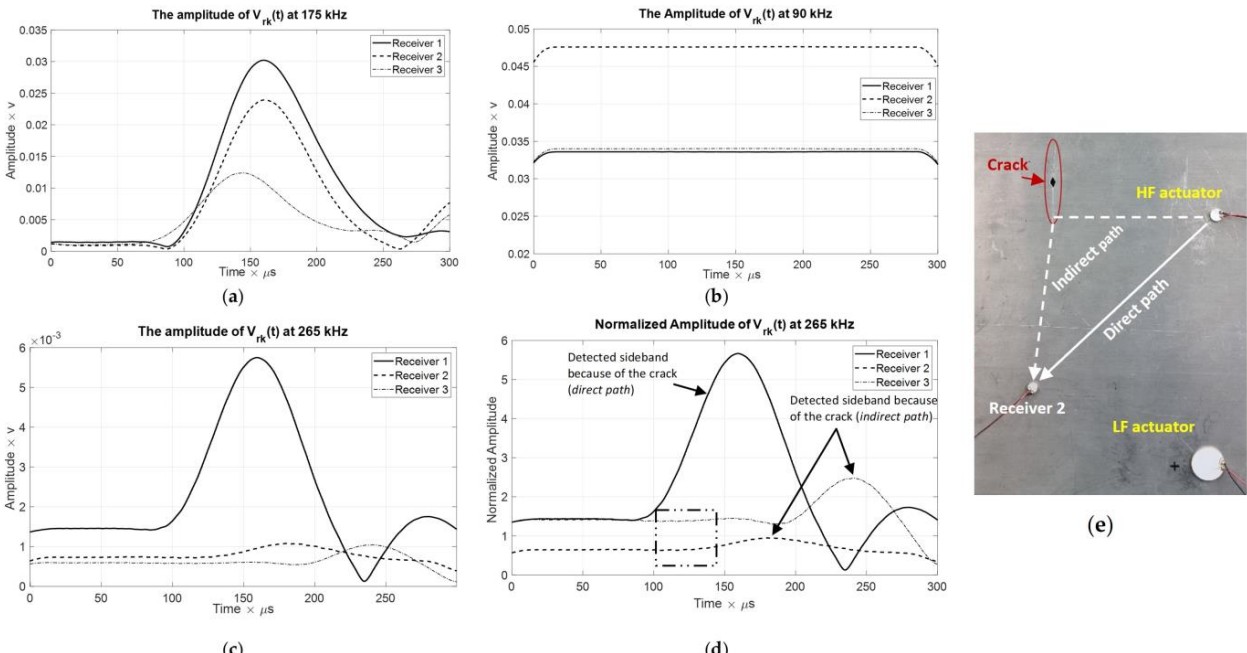

**Figure 13.** (**a**) Amplitude of $V_{rk}(t)$ at 175 kHz; (**b**) amplitude of $V_{rk}(t)$ at 90 kHz; (**c**) amplitude of $V_{rk}(t)$ at 265 kHz; (**d**) normalized amplitude of $V_{rk}(t)$ at 265 kHz; (**e**) direct and indirect signal paths for Receiver 2.

Figure 13a,b illustrates the amplitude of $V_{rk}(t)$ at 175 kHz and 90 kHz for all receivers, respectively. The signal had a constant amplitude at this frequency, due to the continuously excited LF signal. The sideband amplitude for $V_{rk}(t)$ at 265 kHz (175 kHz + 90 kHz) for all receivers is compared in Figure 13c. Observing the curve progressions, it became obvious that the sideband frequency at 265 kHz was highly sensitive to nonlinear defects. However, since the amplitudes of the high- and low-frequency components affected the sideband amplitude (see Equation (3)), the sideband amplitude needed to be normalized for better comparison. $A_\omega$ and $A_\Omega$ represented the respective peak amplitude at 175 kHz and the mean value at 90 kHz. Dividing the sideband amplitude, as shown in Figure 13c, by $A_\omega$ and $A_\Omega$ for each receiver established the normalized sideband amplitude (Figure 13d) [23].

According to the VAM measurement, the signal along Path $P_1$ (as annotated in Figure 8) between the HF actuator and Receiver 1 that ran through the crack delivered a sideband amplitude peak. Receiver 2 received a signal along the direct path (see Figure 13e), with no change in sideband amplitude; however, there was a signal contribution along the indirect path (see Figure 13e) that crossed the crack, thereby delivering a visible sideband amplitude increase. The delay in the sideband amplitude increase was caused by the longer, indirect path. The sideband amplitude of the signal before 100 µs shown in Figure 13d refers to the initial structural nonlinearity since the distributed nonlinearity is contained in the signal and can be quantified.

The VAM algorithm, working with tone burst as an HF signal, allowed us to separate the defect-related nonlinearity from the initial nonlinearity. The black-framed window marked in Figure 13d shows the time interval wherein both frequency components, LF and HF, were received by Receiver 2 and 3; however, their modulation through defect-related nonlinearity had not yet occurred because of the time delay of the signal passing along the indirect path. Obviously, the initial nonlinearity was negligibly small because there was no noticeable increase in the amplitude of the signal within the window. The remaining question posed above concerns how to use this information to localize the defect.

### 4.2. Defect Localization

A test setup for localization is shown in Figure 14a. In this test setup, a 400 ms $S_0$-mode signal at an LF frequency of 69 kHz is excited by two piezoelectric sensors attached on opposite sides of the aluminum plate (Figure 2a). Furthermore, a tone burst HF signal at 175 kHz is excited by another set of two piezoelectric sensors (Figure 2a) after 350 ms of time delay at a different location. The duration of the tone burst signal varies from 50 μs to 150 μs. Simultaneously, the signals are averaged and saved at four receivers positioned in a symmetrical sensor arrangement, as shown in Figure 14a, using a 4-channel RBT2004 digital oscilloscope. It is emphasized that the defect, which is a crack with a length of 30 mm, is positioned outside the arrangement of sensors.

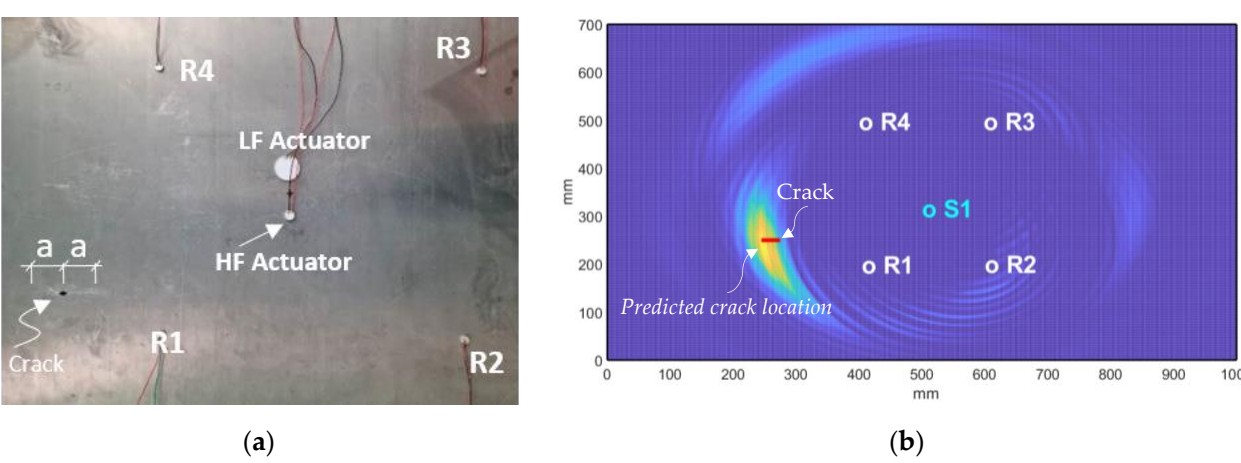

(**a**)                                                                                                     (**b**)

**Figure 14.** (**a**) Defect localization test setup (zoomed in); (**b**) successful defect localization of the crack (a = 30 mm) on a plate of 700 mm × 1000 mm; HF signal duration 100 μs.

In order to cancel out the instrument-related nonlinearity, the normalized SA of $U_{rk}(t)$ is subtracted from the normalized SA of $V_{rk}(t)$. This final SA is used as an input for the DAS imaging method for localization. The TOF from the sender to the defect and from the defect to the receiver (as visualized in Figure 5a) is calculated using Equation (8). The final sideband amplitude $a_k(x,y)$ assigned to this travel time is read out of the SA time histories, as provided in Figure 13d, where $k$ refers to the number of receivers. The equation for the final sideband amplitude of $a_k(x,y)$ is shown in Equation (10). Repeating this procedure for all the coordinates of the grid of the sample plate essentially yields the potential coordinates for the defect, positioned on an ellipse. The ellipses assessed for Receivers R1–R4 are shown in Figure 15a–d, respectively. The yellow ellipse marks the potential locations of the defect. The duration of the HF tone burst was selected to be 100 μs.

$$a_k(x,y) = SA_k(T_{SXR1}(x,y)) \tag{10}$$

A single contour plot of the sideband amplitude $a_k(x,y)$ is created by overlaying the ellipses (Figure 15a–d), thereby revealing the position of the defect on the plate. The final damage imaging is calculated using Equation (11) and is visualized in Figure 14b. The results prove that the localization algorithm locates the area of nonlinearity, coming very close to the actual crack of a = 30 mm.

$$a(x,y) = \prod_{k=1}^{4} a_k(x,y) \tag{11}$$

#### 4.2.1. Tone Burst Duration

Tone burst duration seems to be an important parameter in this process. A too-long tone burst duration essentially causes the overlap of the reflected signals, compromising the signal, while a too-short tone burst duration does not provide enough energy input to

make the nonlinearity recognizable. Hence, the theory is explored as follows: (a) whether a longer signal duration improves the quality of localization results with respect to SA magnitude, and, if this is the case, (b) how short a duration of tone burst can be selected.

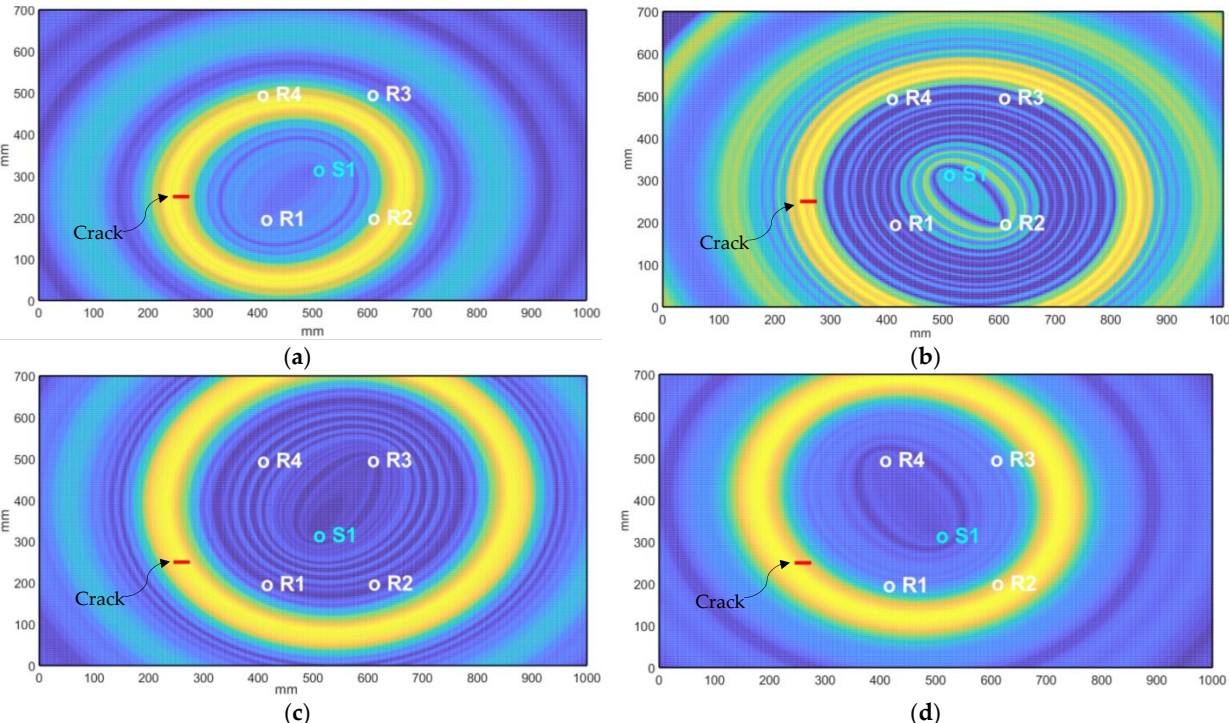

**Figure 15.** Damage imaging for each sender-receiver pair: (**a**) damage imaging for the S1-R1 pair; (**b**) damage imaging for the S1-R2 pair; (**c**) damage imaging for the S1-R3 pair; (**d**) damage imaging for the S1-R4 pair. The crack position is indicated by the red line. Frequencies: LF: 69 kHz; HF: 175 kHz.

The test setup shown in Figure 14 is used for symmetric sensor arrangement. The tone burst durations vary between 50 μs, 75 μs, 125 μs, and 150 μs. Figure 16a–d proves that a longer tone burst duration increases the final SA magnitude, resulting in a more intense DAS marking; however, it is noted that the area of the defect marking spreads more widely with increasing tone burst duration. Therefore, it is concluded that the benefits of longer tone burst durations are limited and a tone burst duration of 50 μs can already provide satisfying results. A shorter tone burst duration diminishes the outcome because of its too-small energy entry and does not yield acceptable results.

### 4.2.2. Symmetric and Non-Symmetric Arrangement of Sensors

Subsequently, a non-symmetric arrangement of Receivers R1–R4 was used to test the localization algorithm, as shown in Figure 17. The HF signal, a 175 kHz tone burst signal with a duration of 100, 125, and 150 μs, was applied. The selected frequency for the LF signal in this test was 76.5 kHz, and the duration of the LF signal was 400 ms. The test outcome clearly shows that a symmetric arrangement of the receivers is not required and does not affect the outcome of the measurement process. Furthermore, it is emphasized that the defect localization algorithm introduced herein was able to detect a defect positioned outside of the sensor arrangement.

### 4.2.3. Localization Error

The measurements reveal a difference between the actual crack location and the predicted crack location, which is defined as the localization error. Equation (12) provides an assessment of the distance between the actual and predicted crack location, using the coordinates of the predicted crack $(x_p, y_p)$ and the actual crack $(x_c, y_c)$. Table 1 lists these distances from the predicted crack locations to the crack tips, as well as the center of the

crack, for use in the measurement scenarios shown in Figure 14, Figure 16, and Figure 17. The last column of Table 1 represents the localization error ratio with respect to the sample dimensions, given as a percentage and assessed via Equation (13). $X_{max}$ and $Y_{max}$ are the sample dimensions of 700 mm and 1000 mm, respectively.

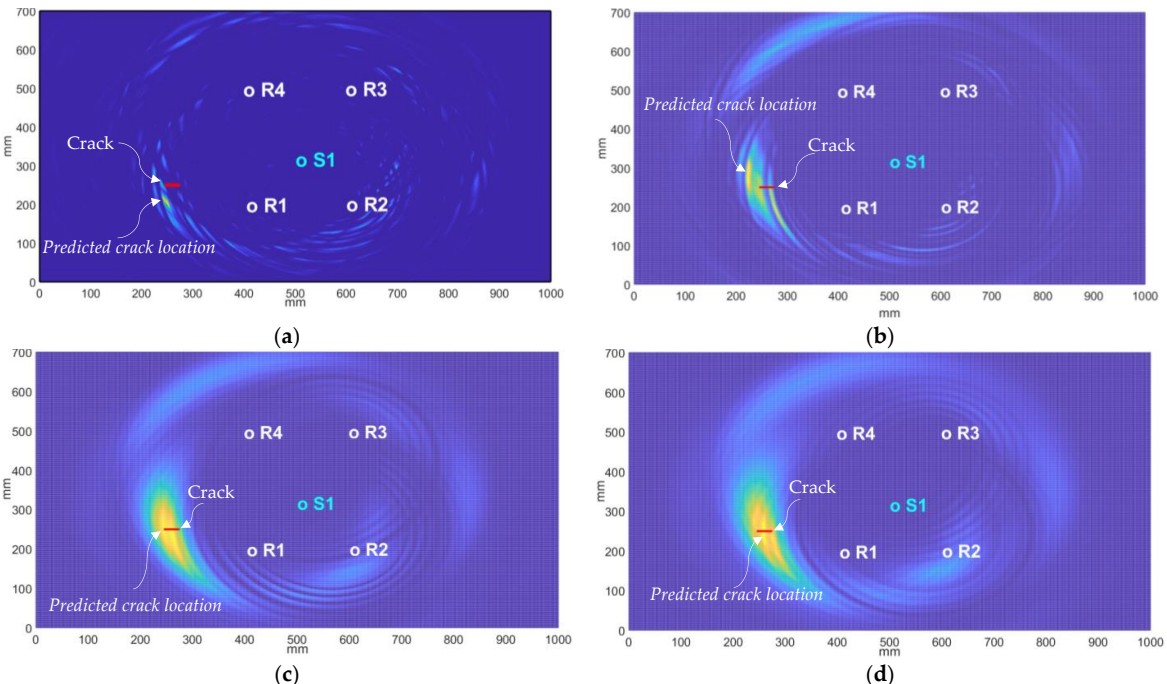

**Figure 16.** Localization results for the: (**a**) 50 μs tone burst; (**b**) 75 μs tone burst; (**c**) 125 μs tone burst and (**d**) 150 μs tone burst. The crack position is indicated by the red line. Frequencies: LF: 69 kHz and HF: 175 kHz.

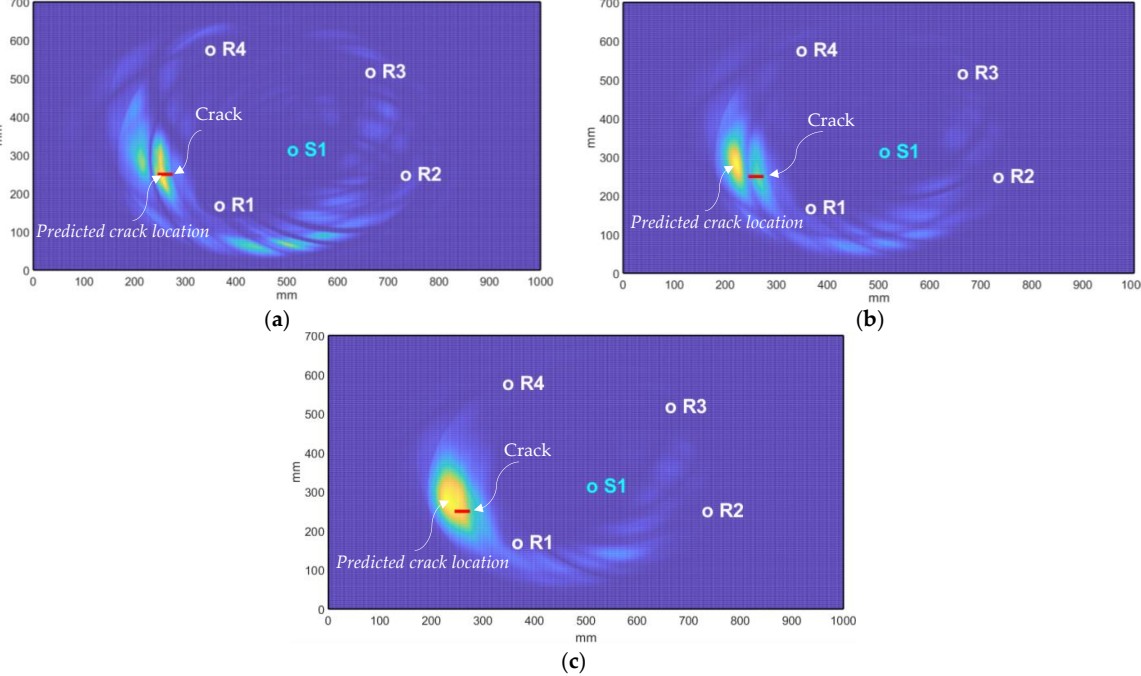

**Figure 17.** Defect localization for the: (**a**) 100 μs, (**b**) 125 μs, and (**c**) 150 μs HF signal. The crack position is indicated by the red line. Frequencies: LF: 76.5 kHz and HF: 175 kHz.

$$Err_{Loc} = \sqrt{(x_p - x_c)^2 + (y_p - y_c)^2} \tag{12}$$

$$Err_{Loc}\% = \sqrt{\left(\frac{x_p - x_c}{X_{max}}\right)^2 + \left(\frac{y_p - y_c}{Y_{max}}\right)^2} \times 100\% \tag{13}$$

**Table 1.** Localization errors.

| ω (kHz) | Ω (kHz) | Tone Burst Duration (µs) | Localization Error (mm) | | | | Localization Error Ratio (%) |
|---|---|---|---|---|---|---|---|
| | | | To the Right Crack Tip | To the Center of the Crack | To the Left Crack Tip | Average | Average |
| 175 | 69 | 50 | 41.05 | 43.01 | 49.65 | 44.57 | 6.12 |
| | | 75 | 37.8 | 48.41 | 60.9 | 49.04 | 5.8 |
| | | 100 | 11.18 | 14.14 | 26.93 | 17.42 | 2.05 |
| | | 125 | 8.06 | 17.88 | 32.02 | 19.32 | 2.14 |
| | | 150 | 17.00 | 2.00 | 13.00 | 10.67 | 1.07 |
| | 76.5 | 100 | 16.28 | 12.65 | 22.47 | 17.13 | 2.12 |
| | | 125 | 41.04 | 50.99 | 62.97 | 51.67 | 6.24 |
| | | 150 | 29.93 | 34.06 | 45.22 | 35.40 | 4.45 |

## 5. Conclusions

After providing a literature review of the current state-of-the-art approaches regarding defect localization, this article focuses on the vibroacoustic modulation (VAM) method, using a high-frequency tone burst signal and a low-frequency continuous signal. A novel methodology is introduced, which combines VAM with the delay and sum damage imaging method to localize the source of a local nonlinearity by tracking the first sideband amplitude. This methodology is proven through experimental testing with a plate-like structure. The methodology successfully localizes a 30-mm crack in a 1.2-mm-thick metal plate-like specimen with 700 mm × 1000 mm side lengths. This study provides an assessment of the localization error.

It is clear that defect localization is possible by applying this methodology, using a minimum number of three piezoelectric sensors. A defect that is positioned outside the arrangement of piezoelectric sensors can still be localized with great accuracy. The methodology allows the researcher to cancel out the nonlinear signal contributions from measuring equipment by subtracting the sideband amplitude of the high-frequency measured signal from the corresponding sideband amplitude of the measured VAM signal. Furthermore, the approach provides the normalized sideband amplitude time histories, whereby defect-related signal measurement can be distinguished from any initial nonlinearities. Finally, the study discusses the necessary tone burst signal durations, in order to ensure successful defect localization.

**Author Contributions:** Conceptualization, M.M.B. and M.R.; methodology, M.M.B.; software, M.M.B.; validation, M.M.B. and M.R.; formal analysis, M.M.B.; investigation, M.M.B.; resources, M.R.; data curation, M.M.B.; writing—original draft preparation, M.M.B.; writing—review and editing, M.R.; supervision, M.R.; project administration, M.M.B.; funding acquisition, M.R. All authors have read and agreed to the published version of the manuscript.

**Funding:** The authors thank the German Research Foundation (DFG) for their financial support of this research (project number: 457416916). Publishing fees are supported by the Funding Program of Open Access Publishing of the Hamburg University of Technology (TUHH) and are appreciated by the authors.

**Institutional Review Board Statement:** Not applicable.

**Informed Consent Statement:** Informed consent was obtained from all subjects involved in the study.

**Data Availability Statement:** The data presented in this study are measured data by piezoelectric sensors and are available on request from the corresponding author.

**Conflicts of Interest:** The authors declare no conflict of interest.

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
