# Peer review of "Defect Localization in Metal Plates Using Vibroacoustic Modulation"

_2813-477X, doi:10.3390/ndt1010002_

Round 1

Reviewer 1 Report

Dear author,

Firstly, I would like to express my gratitude for your research contribution in the field of defect localization using Vibroacoustic Modulation (VAM). Your work provides valuable insights into this area of study. However, while reading your write-up, I noticed some issues regarding the transition between the review articles of previous research and your own contributions.

To enhance the clarity of your paper, it would be beneficial to ensure a smooth transition between the review articles and your work. Clearly demarcating the point where the review articles end and your own research begins will help readers understand the distinct contributions you are making to the field. This could involve providing a clear introductory paragraph or section that sets the context for your specific research objectives and methodology.

(1) By addressing this concern, readers will be able to differentiate between the existing research and the novel aspects you are introducing, enabling them to grasp the unique contributions of your work more effectively.

(2) Also, it is intriguing t that your algorithm allows defect localization with a minimum arrangement of three receiving piezoelectric sensors. Could you explain the rationale behind this choice and discuss any limitations or challenges associated with this minimum sensor configuration?

(3) In your study, you mention that the algorithm successfully localizes defects positioned outside the arrangement of sensors. Could you elaborate on how the algorithm handles such cases and provide insights into the accuracy and reliability of defect localization in such scenarios?

(4) I appreciate that your algorithm demonstrates independence from baseline measurements, which is an attractive feature for practical usage.

However, considering the potential complexities in real-world structures, how robust is the algorithm in terms of accurately localizing defects in situations where the geometry is intricate or the structural characteristics are unconventional?

By addressing these points, you can provide a comprehensive understanding of the algorithm's applicability and shed light on its limitations and potential areas for improvement.

(5) Lastly, coonsidering the potential for further advancements and applications of VAM in defect localization, what are some areas of future research that you believe are worth exploring? Are there any specific challenges or open questions that you believe should be addressed to enhance the effectiveness and reliability of the technique?

Author Response

Dear Reviewer, we appreciate your valuable comments. Please find our response in the attachment.

Reviewer 2 Report

The paper proposes a method for defect localization using two different piezoelectric sensors excited with continuous high-frequency and transient low-frequency lamb waves, which are localized based on the nonlinear interaction of the defect with the lamb wave modes. The content of the paper fits well within the scope of the journal and the manuscript has a clear structure.

    1. Please check the indentation of the whole text. For example, the text below Equation 2 does not need to be indented, while the next paragraphs need to be indented.

2. The paper mentions that the localization of this method has high accuracy and the localization range is much larger than the actual crack area as far as the experimental results are concerned. How to measure the accuracy of the localization results?

3. When setting up the specimen, the author specifically mentions that the through-hole cannot be located on the line of symmetry, what is the purpose of this?

4. It can be seen that the defects in the material actually include cracks and through holes, and they are all located in the same place. Does the through-hole affect the nonlinear detection of cracks?

5. The authors experimentally investigated the effect of many factors on the localization accuracy, such as the duration of short pure tones and the arrangement of sensors. Whether the different crack lengths and sizes have an effect on the experimental results, the authors need to explain in the paper.

fine

Author Response

Dear Reviewer, we appreciate your valuable feedback to the manuscript. Please find our response in the attachment.

Reviewer 3 Report

The paper titled " Defect Localization using Vibroacoustic Modulation" provides a comprehensive overview of defect localization. The topic is of significant importance, and the authors have undertaken a commendable effort in compiling relevant literature and presenting their findings. However, the manuscript requires major revisions to improve clarity, address gaps in the methodology, and strengthen the overall argumentation.

Specific Comments:

Introduction: The introduction provides a clear overview of defect localization papers. However, Studies should be listed from old to new. For example, the paper of Liu et al. [3], which was published in 2016, was mentioned before Qiu et al. [4]'s study, which was published in 2019. Please check this issue again.

The defect localization methodology:

please number the relevant subsections, for example ‘Frequency selection’

Study Limitations: It is important to acknowledge the limitations of the methodology employed. Discuss any potential biases or uncertainties that may affect the interpretation of the results.

1.      P6, L260, "as schematically shown in Fig.2b" - please be more specific on how this can be seen, I can’t see Four receivers.

In many parts of the manuscript, several sentences can be described with one complete sentence. For example, P7, L283-285/P10, L346-366/ P14, L433-434.

P7, L293-294, ‘One way is to cut the signal at a certain time and use it for the localization algorithm’, could you please provide a reference for this statement?!

P9, L340-341, I would strongly suggest showing in a schematic way how to determine the location of a defect by having three receivers and one transmitter.

P9, L343: at -> by

The Specimen: P10, L 384, I think it’s better to delete the table and mention the relevant values in the manuscript.

Measurements, Findings, and Discussion

P12, L 413-414, Figure 9, please label the pictures as you did for Figure 10 or 11.

Figure 10 b and 11b, please add its color bar. Also, please use the same scale for units, for example, if the frequency is written in MHz, it should be written in the same way in the text, not in kHz.

P13, L427-430, this part of the text is not understandable, I would be grateful if it could be reviewed and rewritten.

P14, L440-443, please rewrite these sentences, and why you normalized these data?!

If you want to show results in the same amplitude, is it not better to normalize data with their maximum?!

P14, L445-452, please rewrite this paragraph as well.

Figure 13, please use the appropriate style/type line for plots (dashed lines).

P15, L473-475, please check and rewrite.

P15, L479-480, ‘The time needed for the signal to travel from sender to 479

defect and from defect to receiver’ your mean is Time Of Flight (TOF), if yes, please just simply write TOF!

A very important point, the estimated location is not specified in any of the results (part of Figure 16a). I suggest you display this issue correctly and also present the coordinates and the difference between the estimated value and the actual value in a table.

 Conclusion: The conclusion should provide a concise summary of the key findings and their broader implications. Consider revising the conclusion to ensure it aligns with the stated research objectives and ties together the main points discussed in the paper.

Overall, while the paper addresses an important topic, it requires significant revisions to improve clarity, strengthen the methodology, and enhance the overall argumentation. Addressing the comments mentioned above will substantially enhance the quality and impact of the manuscript.

Choosing appropriate keywords for a paper is essential to increase the discoverability and visibility of your research. Here is some criterion to consider when selecting keywords: identify the main topic, brainstorm related terms, consult existing literature, consider specificity and relevance, check journal guidelines, and so on.

In general, the quality of the English language is good, but it needs to be checked again and some comments mentioned to the authors should be applied.

Author Response

Dear Reviewer, we appreciate your valuable comments which help to increase the quality of the manuscript. Please find our response to your feedback in the attachment.

Reviewer 4 Report

Comments on ndt-2451588

I am very happy to learn that the authors develop a simple but effective technique for defect locating. This research is valuable and may have a high impact on the field. To improve the readability, I have several suggestions for the authors to consider.

1. The terminology should be accurate because I believe that many readers of NDT are engineers. For instance, the words like “hole”, “notch”, “hole notch”, “notch angle”, etc. have specific definitions in various engineering disciplines. To the best of my understanding, the term “hole notch” seems to be a new term coined by the authors, therefore, an explanation should be given. Besides, the statement “The hole notch has sharp notches on two opposite sides. The notch angle is 55°, ... “ is confusing. The so-called “hole notch” actually has “obtuse notches” as well. Why do the authors ignore it?

2. In practical applications, most cases are detecting faults below the surface, not aside. Is the proposed method still valid for subsurface detection?

3. The STFT is a highlight of this study. Several points should be addressed. (a) What type of window is selected? (b) What does “Overlapping number = ????−100” indicate? Does “????“ mean block length and “100” denote sample number? (c) The block length is usually determined by the support of the window function. So what window function do the authors prefer? (d) The statement “FFT points (NFFT) = Sampling frequency/500” is ambiguous. Why that “500” is used? Is it related to Nyquist principle?

4. In my experience, the phase spectrum of STFT provides more insights into the data, particularly for large window durations. The authors don’t have to analyze the phase spectrum but it may be a good idea to mention it somewhere in the discussion/conclusion.

1. Be diligent to check dictionaries for decent wording.

2. Don’t employ the identical sentence pattern or phrase too often and too close in a paragraph.

3. Avoid using passive voice if possible. 

4. Select proper conjunctions to circumvent choppy paragraphs.

Author Response

Dear Reviewer, we appreciate your valuable feedback to our manuscript. Please see our response to your feedback in the attachment.

Round 2

Reviewer 1 Report

Dear author. I agree with all the changes made but it would be nice if a clean copy is produced separately instead of the corrected version with scratch lines.

Author Response

Dear Reviewer, thank you for your efforts. We enjoyed the review process.

Best regards, MM Bazrafkan and M Rutner

Reviewer 3 Report

Dear Authors,

After careful review and consideration, I recommend the final acceptance of the manuscript titled "Defect Localization using Vibroacoustic Modulation" by Mohammad M. Bazrafkan and Marcus Rutner. I have thoroughly evaluated the revised manuscript and the authors have addressed all the concerns raised during the peer-review process, resulting in a significant improvement in the quality and clarity of their work.

The revised manuscript demonstrates substantial contributions to the field and is of high academic merit. The authors have adequately addressed the comments and suggestions made by the reviewers, resulting in a more comprehensive and robust study. The revisions have strengthened the methodology, enhanced the clarity of the results, and improved the overall coherence and argumentation of the manuscript.

Regards,

Author Response

Dear Reviewer, thank you for your efforts and valuable comments. We enjoyed the review process.

Best regards, MM Bazrafkan and M Rutner

Reviewer 4 Report

The manuscript turns out neat. Thanks to the authors for their excellent work.

Just needs a routine pre-publishing editing.

Author Response

(The authors gave the same response as above.)
